# Gallbladder Schwannoma: A Case Report and Literature Review

**DOI:** 10.3390/diagnostics15222827

**Published:** 2025-11-07

**Authors:** Qinyu Liu, Runze Huang, Shujuan Ni, Xin Jin, Xuanci Bai, Lu Wang, Weiping Zhu

**Affiliations:** 1Department of Hepatic Surgery, Fudan University Shanghai Cancer Center, Shanghai Medical College, Fudan University, Shanghai 200032, China; qinyuliu0304@163.com (Q.L.); 23211230016@m.fudan.edu.cn (R.H.); fdzljx@163.com (X.J.); xuancibai116@gmail.com (X.B.); 2Department of Oncology, Shanghai Medical College, Fudan University, Shanghai 200032, China; nsj616@126.com; 3Department of Pathology, Fudan University Shanghai Cancer Center, Fudan University, Shanghai 200032, China

**Keywords:** schwannoma, gallbladder mass, pathogenesis, clinical features, treatment

## Abstract

**Background and Clinical Significance:** Schwannoma is a benign, encapsulated neurogenic neoplasm that originates from Schwann cells of the peripheral nerve sheath. While these tumors may develop in virtually any anatomical location, gallbladder schwannomas are exceptionally rare. **Case Presentation:** A 56-year-old female patient underwent hepatic tumor resection and cholecystectomy following imaging findings suggestive of possible small hepatocellular carcinoma in the right hepatic lobe and biliary cystadenoma. Postoperative pathological examination confirmed that the liver lesion was a lymphoproliferative disorder and that the gallbladder lesion was a classic schwannoma. The patient recovered well with no evidence of disease recurrence during the two-month follow-up. **Conclusions:** Current literature indicates that the pathogenesis of gallbladder schwannomas remains unclear, with no apparent age or gender predilection. These lesions lack distinctive clinical or radiological features, necessitating histopathological confirmation. However, they demonstrate excellent prognosis, with no reported recurrence after complete surgical excision.

## 1. Introduction

Schwannoma, a benign neurogenic neoplasm originating from Schwann cells of the peripheral nerve sheath [1], is typically slow-growing and often presents as an asymptomatic mass. Over 90% of cases are solitary and sporadic [2]. Although schwannomas can arise at any age, incidence peaks between 40 and 60 years, with no clear sex or racial predilection [3,4,5]. These tumors may develop in virtually any part of the body, with common sites including peripheral nerves in the head and neck (particularly the skin and subcutaneous tissues) and flexor surfaces of the extremities. Intra-abdominal schwannomas primarily arise in the retroperitoneum and stomach, whereas gallbladder schwannomas are exceedingly rare [6]. Due to its rarity, there are currently no exact statistics on the incidence rate. Herein, we describe a gallbladder schwannoma occurring alongside hepatic lymphoproliferative lesions and compile a comprehensive review of the literature.

## 2. Case Report

A 56-year-old female patient was admitted to an external hospital on 18 April 2025 for a urinary tract infection. An incidental CT scan at the external hospital revealed a slightly hypodense lesion in the right hepatic dome, suggesting a possible small hepatocellular carcinoma (HCC), and a cystic lesion in the gallbladder fossa, raising suspicion of a cystadenoma. An MRI at the external hospital indicated a possible small HCC in the right hepatic lobe near the diaphragm and a cystic-solid mass in the gallbladder fossa, with unclear boundaries from the gallbladder in some layers, suggesting a liver-origin cystadenoma.

On 22 April 2025, the patient was admitted to Fudan University Shanghai Cancer Center for further evaluation and treatment, with the outpatient diagnosis of “hepatic space-occupying lesion.” Her personal and family medical histories were unremarkable. Laboratory tests were initially conducted for the patient. The patient’s complete blood count and liver function biochemical profiles were all within normal ranges. Among the hepatitis B serology markers, HBsAb, HBeAb, and HBcAb were positive, while HBsAg and HBeAg were negative. All tumor marker levels were normal, with specific values as follows: CA19-9 at 5.04 U/mL (reference range 0–27 U/mL); AFP at 5.01 ng/mL (reference range 0–10.00 ng/mL); CEA at 1.21 ng/mL (reference range 0–5.20 ng/mL); and PIVKA-II at 16.70 ng/mL (reference range 0–21.29 ng/mL). In parallel, imaging studies were conducted for the patient. MRI revealed a nodule in the right hepatic lobe near the diaphragm, measuring approximately 15 mm × 13 mm × 10 mm, with hypointense T1 and slightly hyperintense T2 signals and poorly defined margins (Figure 1A). Contrast-enhanced imaging revealed ring-like arterial enhancement of the lesion, with decreased enhancement on delayed images and hypointensity during the hepatobiliary-specific phase (Figure 1B). No dilation of the intra- or extrahepatic bile ducts was observed. The gallbladder wall showed uneven thickening, and an adjacent irregular cystic-solid mass with hypointense T1 and hyperintense T2 signals was noted, measuring approximately 49 mm × 58 mm × 62 mm, with indistinct borders and unclear separation from the gallbladder in some layers. The mass contained low-signal septations and peripheral solid components (Figure 1D). Contrast-enhanced imaging showed marked enhancement of the septations and solid components, with high signal on DWI and heterogeneous ADC signals (Figure 1E). MRI revealed a small subcapsular lesion in the right hepatic lobe abutting the diaphragm, suspicious for hepatocellular carcinoma. An additional cystic lesion at the gallbladder fossa was favored to arise from the liver, suggestive of a cystadenoma.

Despite the absence of a family history of cancer and normal laboratory and physical examination findings, imaging demonstrated features suspicious for hepatocellular carcinoma in the hepatic lesion and characteristics suggestive of a cystadenoma at the gallbladder. Therefore, surgical resection was recommended.

On 5 May 2025, the patient underwent diagnostic and therapeutic laparoscopic exploration under general anesthesia. Extensive intestinal adhesions were identified, mainly between the intestines and the hepatic and gallbladder surfaces. Considering the large size of the gallbladder tumor, chronic mechanical irritation and localized inflammatory reactions were deemed the major causes of these adhesions. As the adhesions markedly hampered exposure of the hepatic hilum and gallbladder and carried a high risk of enteric fistula during dissection, complete adhesiolysis was performed first. After sufficient exposure of the operative field was achieved, liver tumor resection and cholecystectomy were carried out sequentially. The entire procedure was completed uneventfully without intraoperative complications. Postoperative pathology revealed that the mid-liver tumor was consistent with a hepatic lymphoproliferative disorder (HLPD), measuring 1.4 × 1.2 × 1.5 cm. The gallbladder tumor indicated chronic cholecystitis with cholesterol polyps, while a spindle cell tumor was observed outside the gallbladder wall, measuring 7 × 5.5 × 4.5 cm. Immunohistochemical results for the mid-liver tumor showed a reactive lymphoid hyperplasia pattern: lymphocytes were positive for CD20 (partial), CD3 (partial), CD10 (follicular zone), Bcl-2 (partial), Bcl-6 (follicular zone), and CD5 (partial), while negative for cyclin D1. MNDA (minority+), CD30 (minority+), and C-myc (scattered+) were also observed. Plasma cells were positive for MUM1, kappa, lambda, and IgG4 (minority+). The Ki-67 index was approximately 80% in the follicular zone and 10% in the interfollicular area. CD21 highlighted follicular dendritic cells, and AE1/AE3 marked epithelial components. The gallbladder tumor cells were positive for S-100 and SOX10, negative for CD117 and SMA, and showed a Ki-67 index of 1%. On postoperative pathology, the hepatic lesion proved to be a HLPD, and the gallbladder lesion was a classic schwannoma. Figure 2 shows the histopathological and immunohistochemical findings of the gallbladder lesion.

Postoperatively, the patient received liver-protective, acid-suppressive, anti-infection, and fluid-supportive therapies. She recovered well without complaints and was discharged. During the two-month follow-up, she remained in good condition with no signs of disease recurrence.

## 3. Differential Diagnosis

### 3.1. Preoperative Differential Diagnosis of HLPD Versus HCC

HLPD and HCC can show highly similar appearances on ultrasound, CT, and MRI [7]. For HCC diagnosis, the European Association for the Study of the Liver emphasizes contrast-enhanced imaging with vascular-phase assessment. Typical MRI hallmarks include T1 hypointensity, T2 hyperintensity, arterial phase hyperenhancement, and washout in the portal venous and/or delayed phases, reflecting vascular disturbances during HCC development [8]. However, the diagnostic specificity of both CT and MRI decreases for lesions smaller than 20 mm [9]. The imaging presentation of hepatic lymphoproliferative lesions closely resembles that of HCC [10]. Some reports note that HLPD may show marked restricted diffusion—hyperintense on DWI with low signal on ADC—and may demonstrate internal linear/striated hyperintensity related to dense lymphocyte infiltration [11]. In this patient, MRI did not demonstrate such distinguishing features, making radiologic differentiation from HCC difficult.

Serum AFP is an important biomarker, with approximately 60–70% of HCC patients exhibiting elevated levels [12]. PIVKA-II may also be increased in HCC, whereas CA19-9 and CEA are not typical markers for HCC and are more often associated with cholangiocarcinoma or other gastrointestinal malignancies. By contrast, tumor markers in HLPD are generally within reference ranges [13].

Although preoperative liver biopsy can yield a diagnosis when imaging is indeterminate, it is often avoided because of sampling error and a small but real risk of tumor seeding [14,15]. After comprehensive consideration, surgical intervention was recommended for this patient.

### 3.2. Preoperative Differential Diagnosis of Gallbladder Schwannoma Versus Gallbladder Cystadenoma

According to published cases, gallbladder schwannoma lacks specific imaging or serologic features, and gallbladder biopsy carries higher technical risks; thus, preoperative diagnosis is challenging [16]. Although gallbladder schwannomas usually present as solid masses, larger tumors may undergo cystic degeneration due to insufficient blood supply. At that stage, imaging findings can closely resemble those of gallbladder cystadenoma [17]. On non-contrast CT, these may appear as cystic–solid masses; on contrast-enhanced CT, the cyst wall, internal septations, and any solid nodules typically show mild to moderate enhancement. On MRI, internal septations can be low signal; on post-contrast T1-weighted images, septations and solid components enhance; lesions are often hyperintense on DWI with heterogeneous/low ADC values. In this patient, the imaging features matched the above pattern, making preoperative distinction from a gallbladder cystadenoma difficult.

Serologic tests for both conditions are generally within normal ranges [16,18]. Given the potential for malignant transformation in gallbladder cystadenoma, surgical resection is generally recommended.

## 4. Discussion

### 4.1. Pathogenesis of Gallbladder Schwannoma

#### 4.1.1. NF2 Gene Mutation and Abnormal Molecular Signaling Pathways

According to statistics, more than half of sporadic schwannomas harbor inactivating *NF2* mutations. Neurofibromatosis type 2 (*NF2*) is an autosomal-dominant disorder most often caused by frameshift or nonsense mutations, frequently accompanied by loss of the remaining wild-type allele on chromosome 22. Accumulating evidence indicates that Schwann cell neoplastic transformation is driven by loss-of-function mutations in the *NF2* tumor-suppressor gene, and schwannoma development is causally linked to loss of merlin expression, the growth-inhibitory protein encoded at 22q12.2 [19,20,21,22]. Merlin acts as a tumor suppressor in its open conformation, whereas phosphorylation shifts it to a closed, less active state [23]. Moreover, mutant merlin can form cytoplasmic condensates with IRF3 and TBK1, suppressing the cGAS–STING pathway and thereby impairing antitumor immune responses [24]. Hafez et al. reported a classic case of gallbladder schwannoma arising from an *NF2* gene mutation [20].

Furthermore, it has been demonstrated that the loss of merlin protein in gallbladder schwannoma results in abnormalities in a series of signaling pathways.

##### PI3K/AKT/mTOR Pathway

The PI3K/AKT/mTOR pathway governs essential cellular processes—particularly anabolic metabolism, growth, and survival—and is reported to be upregulated in schwannomas [25]. Mammalian target of rapamycin complex 1 (mTORC1), a principal kinase complex that regulates cell growth, proliferation, motility and survival, functions as a key downstream effector of PI3K/AKT signaling [26]. Notably, mTORC1 inhibitors (e.g., rapamycin) reduce tumor growth in merlin-deficient models, indicating therapeutic potential for subsets of schwannomas [27].

##### Wnt/β-Catenin Signaling

The Wnt/β-catenin pathway is essential for development and disease, and its dysregulation is implicated in multiple malignancies [28,29,30]. Canonical Wnt/β-catenin activation in Schwann cells has been shown to drive schwannoma genesis, accompanied by upregulation of Wnt target genes such as c-Myc and cyclin D1 [31]. This pro-tumorigenic Wnt signaling depends on RAC–PAK–JNK signaling, thereby promoting tumor development [32,33,34].

##### Hippo Pathway

Merlin deficiency in schwannomas leads to Hippo pathway inactivation [35]. This evolutionarily conserved pathway primarily governs cell proliferation, apoptosis, and differentiation, with its dysregulation associated with multiple cancers [36,37].

#### 4.1.2. Germline Mutations

Some patients carry germline mutations associated with the development and progression of schwannomas [38], predisposing Schwann cells throughout the body; consequently, when these genetically susceptible cells sustain a second hit, this can lead to multiple schwannomas, including rare occurrences in the gallbladder. Sadler et al. found that a subset of schwannomas lack *NF2* mutations and implicated loss of heterozygosity at the 9p21.3 locus as an alternative mutational event [39]. In addition, the tumor-suppressor *SMARCB1* at 22q11.2 has been implicated in the pathogenesis of schwannomas in selected cases [40]. Germline mutations in the *SMARCB1* or *LZTR1* genes are detected in 69–86% of familial schwannomatosis cases [41]. Other recurrently altered genes reported in schwannomas include *LATS1*, *LATS2*, *ARID1A*, *ARID1B* and *DDR1* [42,43].

#### 4.1.3. Tumor Microenvironment

The tumor microenvironment (TME) may play a pivotal role in tumor initiation and maintenance, neural infiltration, functional impairment, and neuropathy. Emerging evidence highlights the significant contribution of TME in the pathogenesis and progression of schwannomas [44].

The importance of immune cells within this microenvironment is becoming increasingly apparent. For instance, studies have demonstrated a correlation between tumor-associated macrophage (TAM) infiltration and schwannoma progression [45,46]. Chronic cholecystitis—most commonly associated with cholelithiasis—maintains a chronic inflammatory microenvironment in the gallbladder wall, characterized by dense immune-cell (often lymphoplasmacytic) infiltration [47]. Tumor expansion can create local hypoxia, which activates pro-angiogenic signaling and promotes neovascularization [48]. These rapidly formed, vascular-like structure in the TME express high levels of M-CSF and IL-34, which subsequently regulate the chemotaxis of TAMs. Through the production of growth factors and anti-inflammatory cytokines, these macrophages suppress host immune responses, thereby influencing tumor progression [49,50].

### 4.2. General Characteristics of Gallbladder Schwannoma

#### 4.2.1. Clinical Manifestations of Gallbladder Schwannoma

Schwannoma is a benign neurogenic neoplasm arising from Schwann cells of the peripheral nerve sheath, most commonly affecting the head, neck, and flexor surfaces of the limbs; involvement of the gallbladder is exceedingly rare.

The vast majority of gallbladder schwannomas have an insidious onset and slow growth rate. When the tumor is small, patients are typically asymptomatic; it is often discovered incidentally during abdominal imaging or cholecystectomy performed for other conditions, such as cholelithiasis. As the tumor gradually enlarges or is situated in specific locations, non-specific symptoms such as upper abdominal discomfort, dull pain, or bloating may occur, which can easily be confused with biliary diseases like chronic cholecystitis or cholelithiasis [51]. If the tumor is large or located near the cystic duct or its opening, it may cause cystic duct obstruction, leading to impaired bile drainage, which potentially triggers gallbladder hydrops or even acute cholecystitis.

The following table summarizes the basic clinical information of previously reported cases of gallbladder schwannoma (Table 1).

#### 4.2.2. Imaging Features of Gallbladder Schwannoma

Gallbladder schwannomas lack distinctive imaging characteristics, making preoperative diagnosis extremely challenging. On ultrasound and non-enhanced CT, they typically appear as well-circumscribed lesions with cystic, necrotic, or calcific degeneration. Schwannomas with predominant Antoni A areas exhibit heterogeneous features, while those with Antoni B areas demonstrate a cystic, multilayered structure due to low cellular density [53]. On MRI, schwannomas are typically hypointense on T1-weighted images and uniformly hyperintense on diffusion-weighted and T2-weighted sequences [54]. Contrast-enhanced CT characteristically demonstrates well-circumscribed, encapsulated lesions that may show cystic degeneration [16]. The Antoni A areas, being highly vascularized, show marked enhancement, whereas the Antoni B areas, being less vascularized, exhibit minimal enhancement [53]. Contrast-enhanced ultrasound and CT can assess intratumoral blood flow, aiding in differentiation from malignant tumors [53].

#### 4.2.3. Pathological Features of Gallbladder Schwannoma

Histopathologically, classic schwannomas comprise Antoni A and Antoni B areas. Antoni A-predominant tumors are hypercellular, formed by tightly packed spindle cells with nuclear palisading and Verocay bodies, producing a heterogeneous appearance, whereas Antoni B-predominant tumors are hypocellular with loosely arranged cells and abundant myxoid stroma, often yielding a cystic aspect [55,56].

On immunohistochemical staining, schwannomas show strong positivity for S-100 protein [57]. S-100, although sensitive for Schwann cell lineage, is not specific and can be positive in other neoplasms such as melanoma, neurofibroma, paraganglioma, histiocytoma and clear cell sarcoma [58]. Additionally, SOX10 serves as a diagnostic immunohistochemical marker for peripheral nerve sheath tumors. SOX10 is localized to the nucleus and is expressed in melanocytes, peripheral nerves, breast myoepithelial cells, sweat gland ductal cells, and salivary gland cells, among others [59]. Therefore, the diagnosis of schwannoma should be based on a combination of histopathological examination and immunohistochemical findings.

### 4.3. Histopathological Classification of Schwannomas

Based on the histopathological characteristics of the tumor—that is, the distinct microscopic features such as cellular morphology, architectural patterns, and cellular components (e.g., pigment, glandular structures)—the World Health Organization (WHO) classifies schwannomas into various subtypes [60]. Many of these subtypes exhibit microscopic features that can mimic malignant tumors. Accurate subtyping helps pathologists establish a precise diagnosis, preventing misinterpretation of uncommon benign features as malignant. This supports individualized treatment planning and prognosis assessment, thereby reducing misdiagnosis and overtreatment.

#### 4.3.1. Classic Schwannoma

Conventional schwannoma, the most common histologic subtype, is characterized by alternating Antoni A and Antoni B regions. The tumor frequently shows palisading arrangements of orderly spindle cells with Verocay bodies. Immunohistochemistry typically reveals strong, diffuse S-100 positivity, SOX10 expression, and preserved SMARCB1/INI1 staining.

#### 4.3.2. Ancient Schwannoma

A benign lesion without recurrence potential, featuring degenerative changes including cystic degeneration, hyalinization, calcification, hemorrhage, and necrosis. Contains scattered atypical bizarre nuclei with hyperchromasia and inconspicuous nucleoli. Mitotic figures are rare. May exhibit extensive hyalinization or central ischemic changes [17].

#### 4.3.3. Cellular Schwannoma

Accounts for 2.8–5.2% of benign schwannomas. Predominantly composed of Antoni A areas with minimal/no Antoni B components, Verocay bodies, or palisading patterns. Shows high cellularity with occasional mitoses but no invasive features. Frequently occurs in deep tissues (mediastinum, retroperitoneum, GI tract) and may mimic malignant peripheral nerve sheath tumors [61,62].

#### 4.3.4. Plexiform Schwannoma

Exhibits multinodular growth separated by fibrous septa, often with higher cellularity than classic schwannomas. Predominantly affects skin and superficial soft tissues, particularly in the head/neck region. Histological differentiation from plexiform neurofibroma can be challenging [63].

#### 4.3.5. Epithelioid Schwannoma

Epithelioid schwannoma is composed of round-to-polygonal epithelioid Schwann cells arranged in nests, clusters, or lobules within a myxoid–hyalinized stroma. The tumor cells typically exhibit small, inconspicuous nucleoli and lack mitotic activity. Approximately 40% of cases demonstrate loss of SMARCB1 (INI1) expression, and rare tumors may progress to epithelioid malignant peripheral nerve sheath tumor [64,65].

#### 4.3.6. Microcystic/Reticular Schwannoma

A rare variant predominantly occurring in visceral organs (especially GI submucosa). Demonstrates microcystic/reticular architecture with myxoid/fibromyxoid stroma. Lacks typical Verocay bodies, hyalinization, or foam cells [66,67].

### 4.4. Prognosis of Gallbladder Schwannomas

Gallbladder schwannomas are benign lesions with minimal recurrence risk after complete resection. All reported cases showed no postoperative complications or tumor recurrence during follow-up.

## 5. Conclusions

Gallbladder schwannomas are exceptionally rare and typically lack characteristic clinical manifestations. Patients usually undergo surgical resection due to incidentally detected gallbladder masses on imaging, with definitive diagnosis established through postoperative pathological examination. The recurrence rate following complete surgical excision remains low.

Pathologically, schwannomas are classified into six subtypes; however, reported gallbladder cases predominantly involve classic and ancient schwannoma variants. The diagnosis of gallbladder schwannoma remains challenging, and standardized treatment protocols are currently unavailable. Therefore, elucidating the pathogenesis of gallbladder schwannomas and developing reliable diagnostic and therapeutic strategies represent critical unmet needs in this field.

## Figures and Tables

**Figure 1 diagnostics-15-02827-f001:**
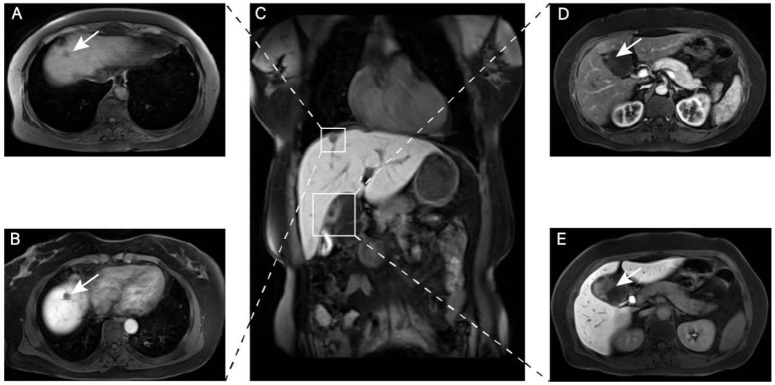
(**A**) A slightly irregular solid lesion at the diaphragmatic surface of the right hepatic lobe. (**B**) The right-lobe, subdiaphragmatic mass demonstrates relatively high signal on the delayed phase of contrast-enhanced imaging. (**C**) Coronal contrast-enhanced images show the subdiaphragmatic solid lesion in the right hepatic lobe and a focal lesion of the gallbladder. (**D**) The gallbladder wall is heterogeneously and mildly thickened with a cystic-solid mass; in focal sections the interface between the mass and the gallbladder wall is indistinct. (**E**) On contrast-enhanced imaging the septations and solid components of the gallbladder wall mass show marked/enhanced contrast uptake.

**Figure 2 diagnostics-15-02827-f002:**
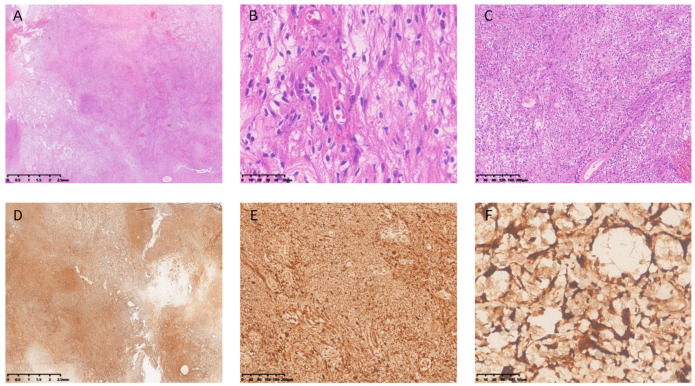
(**A**) Low-power view showing cellular, Antoni A areas and hypocellular, Antoni B areas (×1). (**B**) Medium-power view demonstrating spindle cells in Antoni A areas with focal palisading, while Antoni B areas show cytoplasmic vacuolar change and a loose/myxoid stroma (×10). (**C**) High-power view showing individual tumor cells that are spindle-shaped with scant cytoplasm; no marked nuclear pleomorphism or mitotic figures are identified (×40). (**D**) Low-power view of S-100 immunostaining demonstrating diffuse, strong positivity (×1). (**E**) Medium-power S-100 view showing many spindle cells arranged in nests or fascicles with minimal intercellular stroma; the majority of cells are clearly positive, with staining involving the cytoplasm and perinuclear region (×10). (**F**) High-power S-100 view demonstrating strong cytoplasmic and perinuclear staining of individual cells; cytoplasmic staining is distinct and the background shows a fibrous network and small vessels (×40).

**Table 1 diagnostics-15-02827-t001:** Clinical Characteristics of Previously Reported Cases of Gallbladder Schwannoma.

Case Ref.	Publication Date	Origin	Age (Years)	Sex	Largest Dimension (cm)	Chief Complaint	Characteristics of the Disease	Clinical Course and Prognosis
1 [50]	2010	Japan	58	Male	0.3	Recurrent episodes of right subcostal pain persisting for several years.	Schwannoma of the gallbladder associated with gallstones.	Laparoscopic cholecystectomy.
2 [52]	2014	China	55	Male	2.5	Gallbladder mass found during physical exam.	Primary gallbladder schwannoma.	Underwent laparoscopic cholecystectomy. No signs of recurrence observed at 12-month follow-up.
3 [53]	2016	China	31	Female	5 × 3.5	Intermittent abdominal discomfort with mild bloating and occasional pain for the past 7 years.	Multiple schwannomas located in the porta hepatis, liver parenchyma, and gallbladder.	Underwent open surgical resection of the tumor. No recurrence or complications were observed during the 70-month follow-up period.
4 [54]	2018	China	70	Male	-	Epigastric pain for 3 days with a history of jaundice.	Schwannoma of the gallbladder associated with gallbladder stones and common bile duct stones.	Underwent laparoscopic cholecystectomy. No recurrence detected during 19-month follow-up.
5 [16]	2020	Japan	40	Female	3.7 × 1.7	Gallbladder mass found during physical exam.	Schwannoma of the gallbladder associated with sarcoidosis.	Underwent laparoscopic cholecystectomy with no signs of recurrence at 5-month follow-up.
6 [55]	2020	America	67	Female	4.2	Gallbladder mass found during physical exam.	Schwannoma with degenerative atypia.	Underwent cholecystectomy and gallbladder bed resection. No recurrence observed with good recovery at 2-month follow-up.
7 [56]	2023	China	21	Female	19 × 9	10-week gestation, experiencing epigastric pain for more than 1 month.	Schwannoma of the gallbladder coexisting with pregnancy.	Underwent open cholecystectomy with satisfactory recovery and no postoperative complications.
8 [20]	2024	America	12	Female	17 × 6 × 2	2-year history of biliary colic, featuring moderate-severe dull RUQ pain.	Multiple schwannomas on physical exam, with ocular findings (bilateral ERM, anisometropia, amblyopia, macular scar, eyelid laxity) meeting NF2 diagnostic criteria.	The patient underwent laparoscopic cholecystectomy. Three-month postoperative surveillance revealed no recurrence of gallbladder lesions. With an established diagnosis of Neurofibromatosis Type 2 (NF2), clinical follow-up documented disease progression accompanied by exacerbation of ocular symptoms.

## Data Availability

The original contributions presented in this study are included in the article. Further inquiries can be directed to the corresponding authors.

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
