# Peer review of "Gallbladder Schwannoma: A Case Report and Literature Review"

_diagnostics, 2025, doi:10.3390/diagnostics15222827_

Round 1
Reviewer 1 Report
Comments and Suggestions for Authors
The manuscript presents a very rare case of gallbladder schwannoma occurring concomitantly with hepatic lymphoproliferative lesions, followed by a comprehensive and well-structured literature review on the pathogenesis, characteristics, and subtypes of schwannomas. This is a valuable contribution to the literature given the exceptional rarity of the lesion. Overall, the manuscript is scientifically valid, ethically sound (CARE checklist compliance noted), and its message is clear; however, a few major points must be addressed to enhance its scientific rigor and structural integrity, particularly concerning the case report section and the figures.
Major Flaws Identified
-
Major Flaw in Case Presentation/Data (Clarity and Accuracy): The description of the pathological findings in the liver is a point of confusion and a major flaw in the data presentation. The Abstract (line 20) and Case Report (line 77) state the liver pathology was a lymphoproliferative disorder/reactive lymphoid hyperplasia pattern. However, Figure 2 is introduced as illustrating the pathology, and its caption (lines 90–101) only describes the spindle cell tumor (schwannoma), using terms like "Antoni A," "Antoni B," and S-100/SOX10 staining. The specific microscopic findings and IHC for the hepatic lymphoproliferative disorder are detailed in lines 81–87, but no corresponding figure is provided for the liver lesion (Figure 2a's caption, which is supposed to correspond to the liver tumor, describes Antoni A/B areas, which are features of schwannoma, not lymphoid hyperplasia).
-
Action Required: The authors must correct the caption of Figure 2a to accurately reflect the hepatic lymphoproliferative disorder/reactive lymphoid hyperplasia, or re-label the panels (e.g., if Figure 2a is actually the schwannoma, the labeling is confusing). A separate, clear figure panel must be included (and described in the caption) to visualize the microscopic and IHC features of the hepatic lymphoproliferative lesion. This is critical as the co-occurrence is a key finding.
-
-
Major Flaw in Data Presentation (Figure Captions): The figure captions for Figure 1 and Figure 2 are confusing and misleading.
-
Figure 1 (Imaging): The captions for panels (A) and (C) contradict panel (B). Panel (B) is described as showing "relatively high signal on the delayed phase" for the liver lesion, which is highly suggestive of HCC or an HCC-mimicker, yet the final diagnosis was lymphoproliferative disorder. The authors should explicitly discuss this imaging-pathology mismatch for the liver lesion in the Discussion to avoid a misleading conclusion about the imaging. Furthermore, the caption for Panel (E) appears to describe the same enhancement pattern as the text for the gallbladder mass, potentially making it redundant or poorly distinguished.
-
Figure 2 (Pathology): As noted above, the captions (lines 90–101) seem to exclusively describe the spindle cell tumor (schwannoma) when panels (A) and (B) are introduced as the liver tumor and the gallbladder tumor, respectively. Specifically, the text for (A), (B), (D), (E), and (F) describes the characteristic histology and IHC of a schwannoma (Antoni A/B, S-100 positivity). The authors state the liver lesion was a lymphoproliferative disorder (line 77). This is a major presentation error that must be fixed for the data to be technically accurate. The liver lesion's pathology (lines 81–87) must be clearly represented in a figure.
-
Other Constructive Feedback
-
Scope/Novelty: The case is exceptionally rare and warrants publication. However, the accompanying review on schwannoma pathogenesis (Section 3.1) is extremely broad and not sufficiently focused on the gallbladder or visceral schwannoma. The authors should consider condensing this broad section and focus more on:
-
The possible pathogenetic link between the gallbladder schwannoma and the co-existing hepatic lymphoproliferative lesion. Was a common genetic mutation or pathway investigated? This would significantly enhance the novelty and impact.
-
Pathogenesis relevant to visceral schwannomas specifically.
-
-
Structural Integrity (Conclusion): The Conclusion (lines 280–291) mentions that "reported gallbladder cases predominantly involve classic and ancient schwannoma variants" but the case report does not explicitly state the subtype of the presented gallbladder schwannoma. It is clearly a classic schwannoma based on the Figure 2 description, but the text (lines 78-80) simply calls it a "spindle cell tumor." The authors must explicitly state the subtype in the case report and abstract.
Author Response
We sincerely thank you for taking the time to review our manuscript. Your considerable effort and insightful comments have provided invaluable guidance for improving our work. Each of your suggestions was highly pertinent and has helped us present the research more clearly, strengthen the reasoning, and enhance the overall quality of the paper.
We have thoroughly discussed and addressed every point you raised, and have incorporated corresponding revisions in the revised version (all changes can be viewed using Word's "Track Changes" mode). In addition, we have prepared a detailed point-by-point response explaining how each comment was addressed, with references to the specific locations in the manuscript where revisions were made. Please refer to the attached document for further details.
Once again, we deeply appreciate the time and care you have devoted to reviewing our work, and we are grateful for your support and constructive input.

Reviewer 2 Report
Comments and Suggestions for Authors
The authors reviewed the literature on rare cases of gallbladder schwannoma and whole-body schwannomas.
Recommendations:
1. Were there any abnormalities in the patient's biochemical tests and cancer markers? This should be stated.
2. What is the cause of the patient's bowel adhesion?
3. What type of Schwannoma is your case? This should be stated.
4. What is the incidence of this disease? This should be stated.
Author Response

(The authors gave the same response as above.)

Reviewer 3 Report
Comments and Suggestions for Authors
The manuscript presents an interesting case of an exceptional pathology: gallbladder schwannoma. The reported case is well documented and well illustrated. Furthermore, a literature review addressing the same topic was conducted.
Minor suggestions:
Please consider providing the abnormal bioumoral values of the presented case (if it is the case), tumoral markers including CEA, AFP, CA 19-9, and if upper and lower endoscopy were performed.
The discussion part should also include a differential diagnosis, particularly an imaging one.
Author Response

(The authors gave the same response as above.)

Reviewer 4 Report
Comments and Suggestions for Authors
While gallbladder schwannoma is indeed rare, the manuscript lacks a clear statement of novelty. The case and literature review largely summarize known data without explaining what new insight this case contributes—such as a unique clinical feature, diagnostic pitfall, or histopathologic variant.
The chronology of investigations (CT, MRI, surgery) is somewhat repetitive. Consider presenting it as a clear diagnostic sequence, emphasizing the key clinical decision points rather than listing redundant details.
The case presentation could better describe symptoms, laboratory results (especially liver enzymes, bilirubin, or tumor markers), and differential diagnoses that were considered preoperatively. This would make the reader appreciate why malignancy was suspected.
The MRI figures are well described, but captions are overly technical and repetitive. Including side-by-side comparative imaging (CT vs MRI) or annotated arrows to key features would strengthen clarity for general readers.
The histological features are well illustrated but could be better connected to diagnostic reasoning—how the diagnosis of schwannoma (vs GIST or leiomyoma) was confirmed immunohistochemically. A brief differential diagnosis paragraph would add value.The detailed genetic and signaling pathway descriptions (NF2, SOX10, PI3K/mTOR, etc.) are scientifically sound but disproportionate to the clinical relevance of this case. A concise summary focusing on gallbladder-specific implications would make the discussion more focused.
The authors could enhance clinical applicability by proposing a diagnostic approach for suspected gallbladder schwannoma—highlighting imaging clues, differential diagnoses, and when to consider histology or IHC.
Author Response

(The authors gave the same response as above.)

Round 2
Reviewer 1 Report
Comments and Suggestions for Authors
I am fully satisfied with the authors' responses and the resulting revisions. The manuscript is now well-prepared for publication in "Diagnostics" and serves as an important educational piece for clinicians. The authors' diligence in addressing every point, no matter how small, reflects a commitment to scientific rigor.
Author Response
We extend our heartfelt thanks for your valuable feedback and the considerable effort dedicated to reviewing our manuscript. Your insightful comments were essential for enhancing the paper's quality, scientific rigor, and educational impact.
We are pleased that you find the revised version satisfactory and appreciate your recognition of its educational merit. Your input has been immensely helpful and encourages our ongoing commitment to high-quality research.
Thank you again for your constructive guidance.
Reviewer 2 Report
Comments and Suggestions for Authors
The authors reviewed the literature on rare cases of gallbladder schwannoma and whole-body schwannomas.
They responded well to my suggestions and criticisms. I would like to thank them.
Suggestions:
1. They responded to my initial recommendation regarding intestinal adhesions, but I couldn't find this explanation in the article. For such a rare condition, this valuable information should also be included in the article.
Author Response
We sincerely thank you for pointing out the necessity of explaining the cause of the patient's intestinal adhesions. We fully agree with this comment and deeply regret our failure to adequately elaborate on this point in the original manuscript.
Therefore, after consulting relevant literature and conducting in-depth discussions with several clinical experts in hepatobiliary diseases, we have carefully revised the section in the article concerning the performance of adhesiolysis and have provided an explanation for the cause of intestinal adhesions in this patient (Lines 95–104 in the revised manuscript, or Lines 89–98 with tracking turned off).
We hope that these modifications meet your requirements, and we thank you once again for your valuable guidance.